# Aberrant Glycosylation as Immune Therapeutic Targets for Solid Tumors

**DOI:** 10.3390/cancers15143536

**Published:** 2023-07-08

**Authors:** Yasuyuki Matsumoto, Tongzhong Ju

**Affiliations:** Office of Biotechnology Products, Center for Drug Evaluation and Research, The U.S. Food and Drug Administration, Silver Spring, MD 20993, USA

**Keywords:** glycosylation, glycans, glycoproteins, tumor-associated carbohydrate antigens (TACAs), cancer, immunotherapy, CART, GD2, Tn antigen, glyco-vaccine

## Abstract

**Simple Summary:**

Glycosylation is one of the most pivotal post-translational modifications on all types of biomolecules for the formation of glycoproteins, glycolipids, and glycoRNAs in a tissue-type specific manner. Normal glycans participate in biological events such as development, metabolism, differentiation, and immunity in mammalian cells. In cancers, the altered glycosylation, known as tumor-associated carbohydrate antigens (TACAs), play important roles in facilitating tumor formation, progression, and metastasis. Recent studies also reveal novel roles of TACAs in facilitating the vulnerable tumor microenvironment through the interaction of glycan binding receptors expressed on immune cells. TACAs are specific and expressed on different cell surface molecules in a cancer type-specific manner. Thereby, TACAs are potential tumor glyco-biomarkers, glycoimmune checkpoints, and therapeutics. In this review, we summarize the established and most promising immunotherapy-targeting TACAs such as monoclonal antibody therapy including glycoengineered humanized antibody and antibody-drug conjugates (ADC), chimeric antigen receptor T-cell (CART) therapy, and vaccination in recent clinical trials.

**Abstract:**

Glycosylation occurs at all major types of biomolecules, including proteins, lipids, and RNAs to form glycoproteins, glycolipids, and glycoRNAs in mammalian cells, respectively. The carbohydrate moiety, known as glycans on glycoproteins and glycolipids, is diverse in their compositions and structures. Normal cells have their unique array of glycans or glycome which play pivotal roles in many biological processes. The glycan structures in cancer cells, however, are often altered, some having unique structures which are termed as tumor-associated carbohydrate antigens (TACAs). TACAs as tumor biomarkers are glycan epitopes themselves, or glycoconjugates. Some of those TACAs serve as tumor glyco-biomarkers in clinical practice, while others are the immune therapeutic targets for treatment of cancers. A monoclonal antibody (mAb) to GD2, an intermediate of sialic-acid containing glycosphingolipids, is an example of FDA-approved immune therapy for neuroblastoma indication in young adults and many others. Strategies for targeting the aberrant glycans are currently under development, and some have proceeded to clinical trials. In this review, we summarize the currently established and most promising aberrant glycosylation as therapeutic targets for solid tumors.

## 1. Introduction

Glycosylation is one of the most pivotal post-translational modifications (PTMs) on all types of biomolecules, including proteins, lipids, and ribonucleic acids such as RNAs to form glycoproteins (including mucins), glycolipids, and glycoRNAs, collectively known as glycoconjugates [1,2,3,4]. Glycans, which are the carbohydrate moieties in glycoconjugates, have different structures in a cell or tissue-type specific manner to compass the glycome to act at biological events such as development, metabolism, differentiation, and immunity [5,6,7,8,9,10]. The altered glycosylation with unique structures in cancers, known as tumor-associated carbohydrate antigens (TACAs), plays crucial roles in facilitating tumor formation, progression, and even metastasis [11,12,13,14,15,16,17]. TACAs are often expressed in a cancer type-specific manner (i.e., CA 19-9), which is a serum biomarker of digestive carcinomas used in clinic [18]. TACAs are potentially useful as tumor glyco-biomarkers in clinical practice, as well as the therapeutic targets for treatment of cancers [15,19,20].

In over four decades of history in glyco-oncology, TACAs themselves, and also TACAs on specific proteins and lipids, have been recognized as cancer antigens at high rankings (i.e., MUC1 in a second rank) based on comprehensive criteria such as therapeutic value, immunogenicity, specificity, and number of epitopes, etc. [21]. There is a list of well-characterized TACAs in many types of human carcinomas: Truncated mucin-type *O*-GalNAc glycans, Tn and STn antigens; Sialyl Lewis-related antigens, SLeA, SLeX, and SLeC; Glycosphingolipids, GM3, GD3, GD2, fucosyl-GM1, and Globo H (Figure 1). Additionally, serum glycoproteins, such as CA 15-3 (MUC1), CA 125 (MUC16), CA 72-4 (MUC1), CEA (CEACAM5), and PSA, have been developed as candidates of diagnostic biomarkers [22,23,24,25,26]. As highlighted in TACA-based immunotherapy, several anti-GD2 monoclonal antibodies have been developed and approved by the U.S. Food and Drug Administration (FDA) and European Medicines Agency (EMA) for neuroblastomas in young adult patients, and other TACAs are under development as tumor targets with promising anti-tumoral effects in preclinical/clinical studies. In addition, some TACAs are currently being understood as a novel glycoimmune checkpoint (TACA-glycan binding lectin receptor (GBR) axis) to facilitate immunosurveillance evasion by generating the tumor microenvironment (TME) [17,20,27,28,29,30,31,32,33]. Here, we summarize and discuss well-established and the most promising therapeutic targets among TACAs for monoclonal antibody (mAb) immunotherapy, antibody-drug conjugates (ADCs), chimeric antigen receptor T-cell (CART) therapy, bispecific Ab (BsAb), bispecific T-cell engagers (BiTEs), and vaccination therapy in recent clinical trials.

## 2. TACA-Targeted Immunotherapy

### 2.1. Tn and STn Antigens as Pan-Carcinoma TACAs

Among highly relevant TACAs, the Tn antigen (GalNAcα1-*O*-Ser/Thr/Tyr, Figure 1) and its sialylated version of the Tn antigen (STn, Neu5Acα2-6GalNAcα1-*O*-Ser/Thr, Figure 1) are widely expressed in a majority of human carcinomas originated from gastrointestines, pancreas, lung, breast, prostate, bladder, and ovary [13,14]. Tn and STn antigens, categorized as pan-carcinoma antigens, are truncated *O*-GalNAc or mucin-type *O*-glycans and associated with poor prognosis and cancer progression such as early carcinogenesis, progression, metastasis by suppression of immunosurveillance [34,35,36,37,38,39,40,41,42]. In addition to their expression on the plasma membrane of tumor cells, a variety of Tn/STn-carrying glycoproteins, such as mucins, CA 15-3 (MUC1), CA 125 (MUC16), and CA 72-4 (MUC1), are secreted in circulation in patients with gynecologic cancer [43,44,45,46,47,48,49], digestive cancer [50], gastric cancer [51,52], pancreatic cancer [53], and endometrial cancer [54], and the titers of those serum glycoproteins correlate with cancer malignancies [55,56]. Thus, they have been developed as diagnostic biomarkers in a cancer type-specific manner. While those diagnostics are evaluated with specific mAbs that generally recognize both carbohydrates and surrounding peptides, it would be quite challenging to generate mAbs against such shortened glycan epitopes, particularly the Tn antigen, which monosaccharide, α-linked GalNAc is less immunogenic and may not be processed by MHC-II T cell-dependent immune response [57,58,59]. Another complexity is that anti-Tn mAbs may cross-react with α-linked GalNAc-terminating structures, such as blood group A (BGA) and Forssman-related antigens [60,61], resulting in false positivity in serum diagnosis. Additionally, human IgA1 in circulation carries the Tn antigen in its hinge region [62,63,64], which may also interfere with the diagnostic specificity. Nevertheless, specific anti-Tn mAbs are widely under investigation as valuable tools for cancer diagnosis and therapeutics.

Approximately 30 different anti-Tn/STn mAbs have been developed and reported [65,66,67,68,69,70,71,72,73,74,75,76,77,78,79,80,81,82,83,84,85,86,87,88,89,90,91,92,93,94], and several mAbs show the therapeutic potential in preclinical study [68,77,78,79,81,91,92,94,95,96,97,98,99]. Specifically, in the 1990s–2000s, several radioisotope-labeled anti-STn mAbs were investigated in clinical trials. ^177^Lu-CC49 mAb radioimmunotherapy was studied in chemotherapy-resistant ovarian cancer patients, resulting in prolonged disease-free survival in Phase I/II clinical trials [100]. Later on, a combination therapy of ^177^Lu-CC49 mAb with IFNα and paclitaxel was tested in ovarian cancer patients who failed a standard therapy, as IFNα co-treatment expects to enhance intraperitoneal radioimmunochemotherapy, showing that the drug was generally tolerated in Phase I clinical trial (NCT00002734) [101]. Similarly, ^131^I-CC49 mAb with IFNα radioimmunotherapy was tested in hormone-resistant metastatic prostate cancers, showing better anti-tumor effects as compared to mAb treatment alone (NCT00025532) [102]. ^131^I-CC49 mAb monotherapy was also tested for patients with recurrent or metastatic colorectal cancer in Phase I trial (NCT00023933). None of those trials, however, have yet entered in Phase III trial. It is worth noting that, in addition to the STn antigen, CC49 mAb also recognizes the Neu5Acα2-6GalNAc moiety within sialyl Core 1 [Galβ1-3(Neu5Acα2-6)GalNAc], or disialyl Core 1 [Neu5Acα2-3Galβ1-3(Neu5Acα2-6)GalNAc], which are present in the normal cellular *O*-glycome [103]. This may raise ‘off-target’ concerns associated with adverse events.

Recently, studies showed that antibody-drug conjugates (ADCs) with anti-Tn mAb [104] and anti-STn mAbs [92,105,106] exhibited anti-tumor activity in preclinical settings. Furthermore, 5E5, one of the anti-Tn mAb that specifically recognizes Tn-carrying MUC1 (Tn^+^MUC1) [76], was employed on chimeric antigen receptor T-cell (CART), and demonstrated specific cytotoxicity to leukemic and pancreatic cancers with Tn^+^MUC1 expression using an in vivo xenografted mouse model [107]. This Tn^+^MUC1-CART therapy is currently undergoing in clinical Phase I study for the patients with Tn^+^MUC1 positive advanced cancers (NCT04025216). The other related CART therapies-targeting MUC1 are also being tested in clinical trials (discussed in Section 2.4). In the meantime, anti-STn antibody, such as TAG72 (CA 72-4, i.e., STn-carrying MUC1), was initially applied in a first generation of CART platform over three decades, however, TAG72-CART therapy failed in Phase I study as a result of limited T cell persistence and lack of precise humanization on scFv of TAG72 [95,108,109]. Recently, humanized TAG72 (huCC49) was applied in a second generation of CART, showing a significantly prolonged survival rate in an ovarian cancer xenografted mouse model [110], and TAG72-CART therapy is currently entering in clinical Phase I trial for treatment of patients with platinum resistant epithelial ovarian cancer (NCT05225363).

Interestingly, elevated titers of anti-Tn/STn antibodies were observed in sera from patients with gastrointestinal cancer [111], prostate cancer [112], and breast cancer [113]. Although those antibodies are also found in healthy donors at some levels due to infections of microorganisms and parasites [114,115,116,117,118], elevation of those antibodies would be good indicators of diagnosis/prognosis in cancers. Tn and STn antigens are not solely biomarkers, they play roles in immune tolerance and anti-inflammatory response in which macrophage galactose-type lectin (MGL, also known as CLEC10A) on macrophages and dendric cells (DCs) binds to the Tn antigen, facilitating the generation of hostile tumor microenvironment (TME) within and even around tumor tissues (Figure 2) [28,119,120,121,122,123,124,125]. Furthermore, the STn antigens on mucins overexpressed in cancer cells create an immunosuppressive environment in which sialic acid-binding immunoglobulin (Ig)-like lectins (i.e., SIGLECs) on most of innate immune cells and NK cells bind to the STn antigen, leading to immunosuppression and evasion of immunosurveillance (Figure 2) [29,30,31,126,127,128]. Studies also showed that Tn and STn antigens trigger epithelial-mesenchymal transition (EMT), invasion, metastasis, and immune surveillance [129,130,131]. Therefore, vaccination with the Tn/STn antigens-carrying immunogens has been proposed and studied in clinical trials.

Notably, a recombinant chimeric anti-Tn human IgG1 mAb, Remab6, was generated by the de novo sequencing of a murine anti-Tn IgM mAb, CA3638 (BaGs6), which was originally generated by an immunization of mouse with Tn^+^ erythrocytes from patients with Tn syndrome [74,86]. The anti-Tn IgM mAb (CA3638) was shown to be specific to the Tn antigen in human cancer cell lines [132], human carcinoma tissue sections [133], mouse tissues with *c1galt1c1* (COSMC) gene knockout [134,135,136,137,138,139], or *c1galt1* (T-synthase) gene knockout [140]. The epitope requirement was determined to be two consecutive Tn antigens or two Tn antigens in a conformationally proximity regardless of the peptide sequences [141]. The Remab6 showed high specificity to human carcinomas including gastrointestinal, breast, prostate, ovarian, pancreatic tissues of tumor origin, but not normal tissues [86,142]. In therapeutic point of view, the core fucosylation of *N*-glycans at N297 in the CH2 domain on human IgG negatively regulates antibody-dependent cellular cytotoxicity (ADCC) by interfering with IgG-Fc receptor interaction on effector cells [143,144,145]. The glycoengineered version of Remab6, named Remab6-AF (aFucosylated), showed a potent ADCC to Tn^+^ human tumor cells in both in vitro and in vivo settings [99]. Importantly, several anti-Tn/STn mAbs mostly bind Tn/STn-carrying mucins, which would not induce complement-dependent cytotoxicity (CDC) activity in many cases [146]. Remab6, however, can target multiple Tn^+^ glycoproteins other than mucins, and share some Tn^+^ glycoproteins that are recognized by MGL receptor [147], suggesting that it would overcome one of the big hurdles of immunotherapy targeting the Tn antigen and the TME in human carcinomas.

In summary, although some of anti-Tn mAbs are not highly specific to the Tn antigens (GalNAc-α1-*O*-S/T/Y) often expressed on tumor cells, they can still be valuable for cancer diagnosis if proper controls are included. Furthermore, the anti-Tn immunotherapies hold great potentials since targeting the Tn antigens is not solely attacking the tumor cells but also activating local immune cells to overcome the TME.

### 2.2. CA 19-9 in Pancreatic and Digestive Carcinomas

CA 19-9, terms sialyl Lewis A (SLeA, Figure 1), is one of the most classic TACAs and clinically used as a diagnostic serum biomarker for detecting pancreatic carcinoma over four decades [16,148]. Magnani et al. initially described the putative structure of SLeA as a glycolipid of monosialoganglioside using a monoclonal antibody immunized with a human colorectal carcinoma cell line, SW1116 [149,150]. Later on, the epitopes of SLeA were also found on both *N*- and *O*-glycans on mostly mucins, and apolipoproteins in sera of gastrointestinal and pancreatic carcinoma patients [151,152,153]. It has been shown that SLeA, in general, contributes to vessel invasion and extravasation-mediated platelets and leukocyte attachment to endovascular tissues [12,154,155,156,157,158]. CA 19-9 is elevated in sera of patients with gastrointestinal and pancreatic cancers [25,50,52,159,160,161,162,163,164]. However, a series of studies demonstrates that diagnostic test with CA 19-9 may not be specific for those cancer patients since CA 19-9 levels in sera are also found to be elevated in patients with pancreatitis, cirrhosis, and biliary obstruction, causing a false positivity in many cases [149,165,166]. The other study demonstrates that SLeX (see also Figure 1), which is an isomer of SLeA, is elevated in sera of patients with pancreatic cancer that are SLeA-negative [167]. Currently, CA 19-9 is only used for detecting pancreatic cancer and monitoring in the course of treatment of patients with gastrointestinal carcinomas [168,169].

Several mAbs to CA 19-9 have been developed and evaluated in detecting pancreatic carcinomas [170]. Characterization of those mAbs on glycan arrays revealed that some mAbs show a broad specificity beyond SLeA, which causes cross-reactivity with sialyl Lewis C (SLeC, also known as DU-PAN-2, see also Figure 1) that lacks fucose on SLeA [171,172]. Of note, the individuals who do not express α1,3-Fucosyltransferase-3 (FucT-3) that is a responsible enzyme to form Lewis A structure, known as Lewis-negative, even suffered from pancreatic cancer [170,173,174]. Approximately, 20% of Caucasian and African American populations are Lewis-negative due to genotypic variants and lower activity of FucT-3 [173], which led to the false negative diagnosis of CA 19-9 in patients [168,175]. A few studies demonstrated the elevation of serum levels of SLeC in Lewis-negative patients with pancreatic cancer [174], and also showed that serum levels of SLeC was higher than CA 19-9 in patients with digestive cancer [176].

In recent study, a highly specific mAb targeting CA 19-9, MVT-5873, which is a human IgG1 derived from lymphocytes in a breast carcinoma patient immunized with SLeA-KLH plus saponin adjuvant vaccine, was developed and is being tested for treatment of patients with pancreatic cancer and metastatic colorectal cancer in Phase I and II trials (NCT02672917, and NCT03801915, respectively) [177,178]. This human IgG1 is also tested as an experimental imaging agent in PET/CT scans for diagnosis of pancreatic cancer in Phase I trials (NCT04883775, and NCT05737615). The other defined antibody, CH129 that targets sialyl-di-Lewis A, was conjugated with MMAE (Monomethyl auristatin E) to form ADC version of a chimeric human IgG1, and showed anti-tumoral effects in an in vivo mouse model [179]. Interestingly, a highly specific scFv against CA 19-9, RA9-23 was developed by a yeast surface display platform with an 1116-NS-19-9 scFv backbone [149], which may be useful tools of selection for effective anti-glycan antibodies with diagnostic and therapeutic potentials [180].

### 2.3. GD2 in Neuroblastoma and Glioma

GD2 (Figure 1) is an intermediate member of acidic glycosphingolipids (GSLs) overexpressed in melanoma, glioma, neuroblastoma, and small cell lung cancer (SCLC) [9,11,12,181,182,183,184]. The expression of GD2 is barely seen in normal tissues, except in brain and skin with weak expression levels, as GD2 is an intermediate product in the ganglioside biosynthesis [185]. Some GSLs act as immune-suppressors and harness oncogenic signaling pathways upon clustering membrane-bound molecules in microdomains to induce cell growth, adhesion, and invasive activity in melanoma, neuroblastoma, glioma, SCLC, and osteosarcoma [9,11,186,187,188,189,190,191,192,193,194,195,196,197]. A recent study suggests that GD2 interacts with SIGLEC7 on innate immune cells to suppress local immunity, thus promoting tumor cell evasion from immunosurveillance (Figure 2) [198,199]. Therefore, GD2 is recognized as TACAs for a variety of solid tumors and a target for cancer therapy [200].

Several mAbs targeting GD2 have been developed and showed anti-tumoral effects with CDC and ADCC against neuroblastoma and lymphoma in vitro and in vivo [201,202,203,204,205,206]. Other than antibody-mediated cytotoxicity to cancers, one of the anti-GD2 mAbs, 220-51, exhibited suppression of cell growth and induction of anoikis in SCLC in vitro [207,208].

Dinutuximab (Unituxin^TM^ [209]), a chimeric human anti-GD2 mAb, was approved by the FDA for treatment of patients with high-risk neuroblastoma in 2015 (NCT00026312) [210,211]. Dinutuximab is originally derived from 14.18 mouse hybridoma and expressed in SP2/0 cells by the Reisfeld group [202]. Recent studies reported that Dinutuximab-treated patients with neuroblastoma relapsed due to an adrenergic-to-mesenchymal transition (AMT), leading to low expression levels of GD2 [212]. It was found that treatment with EZH2 inhibitor (Tazemetostat, an FDA-approved drug to follicular lymphoma, NCT01897571) blocks AMT and induces re-expression of GD2 in neuroblastoma, suggesting that combination treatment of two drugs would be clinically beneficial for children with high-risk neuroblastoma [212]. In another study, Dinutuximab was reported to suppress cell growth in GD2^+^ cell lines of neuroblastoma, osteosarcoma, and SCLC and block an immunosuppressive axis of GD2-SIGLEC7 in macrophages, enhancing phagocytotic activity with no obvious neurotoxic side-effects in an in vivo mouse model [198]. These phagocytotic effects are significantly synergized with co-treatment of anti-CD47 mAb (Magrolimab) that inhibits SIRPα-mediated ‘don’t eat me’ signaling in macrophages. A combination of those two therapeutic drugs (Dinutuximab/Magrolimab) is now being tested in patients with relapsed or refractory neuroblastoma or relapsed osteosarcoma in Phase I trial (NCT04751383). In other anti-GD2 immunotherapeutic strategies, a fusion protein with a humanized version of anti-GD2 mAb (hu14.18)-linked to human interleukin 2 (IL2) (EMD273063) was developed and tested for patients with GD2^+^ tumors such as neuroblastoma and melanoma in Phase I/II trials (NCT00590824, and NCT00082758) [213,214,215,216]. Dinutuximab beta is the dinutuximab manufactured in Chinese hamster ovary (CHO) cell line, was developed and approved in Europe, and is currently in clinical trials in the US (NCT05558280, NCT05272371, NCT04253015, and NCT05080790). Currently, a combination of Dinutuximab beta and intravenous IL2 administration is also being tested for high-risk neuroblastoma patients in Phase III trial (NCT01704716) [217].

Naxitamab is another humanized therapeutic anti-GD2 mAb (hu3F8) originally derived from 3F8 mouse hybridoma [201]. The combination of Naxitamab with GM-CSF immunotherapy was approved by the FDA in 2020 for patients with relapsed or refractory neuroblastoma (NCT03363373, and NCT01757626) [218,219,220]. Furthermore, a bispecific antibody (BsAb) targeting GD2 and CD3 (Nivatrotamab, hu3F8 x huOKT3 BsAb [221]) has entered in Phase I/II trials for patients with metastatic SCLC, neuroblastoma, and osteosarcoma (NCT03860207, and NCT04750239). The trials were terminated due to business priorities.

For potential CART therapy targeting GD2, two GD2-CART platforms were developed for treating neuroblastoma and glioma in an in vivo mouse model [222,223], and recently a second generation of GD2-CART therapy with anti-GD2 scFv, originally derived from 14g2a [224] for patients with diffuse intrinsic pontine glioma (DIPG), or osteosarcoma/neuroblastoma has entered in Phase I trials (NCT04196413, and NCT04539366, respectively) [225]. Along the same line, a second generation of GD2-CART platform with anti-GD2 scFv, huK666, originally derived from KM8138 [204] with an inducible caspase 9 suicidal gene cassette (iCasp9), has been developed [226]. This platform may have better control of off-target effects on central and peripheral nervous system that weakly expresses GD2, and is being tested in Phase I trial (NCT02761915) [227]. The fourth generation of GD2-CART (4SCAR-GD2, hu3F8 scFv-fused with a domain with CD3-zeta, 4-1BB, CD28, and inducible caspase 9) for pediatric patients with neuroblastoma demonstrated anti-tumoral effects with less toxicities such as cytokine release syndrome (CRS) and neuropathic pain in a Phase I study (NCT02765243) [228]. In other GD2-based CART therapies, a gene of IL7 receptor (C7R) is added on a second generation of CART (C7R-GD2-CART) and is now being tested for patients with GD2^+^ brain tumor in Phase I trial (NCT04099797). CART cells with an introduction of IL7 receptor is expected to survive for a longer period of time due to stimulation by IL7.

### 2.4. Targeting Neoantigens in Carcinomas

A class of most well-known neoantigens is mucins (i.e., MUC1, MUC16), which is observed in both membrane-bound and secretory forms in patients originated from gastrointestines, breast, pancreas, and ovary [21,229,230]. CA 15-3 (MUC1) and CA 125 (MUC16) are serum biomarkers for patients with breast, lung, pancreas, ovarian, and prostate cancers [231,232,233], and their titers and glycoforms were evaluated in several diagnostic studies (See more details in Section 2.1). Studies showed that those oncogenic mucins are highly expressed and heavily glycosylated with shortened/truncated *O*-glycans such as Tn/STn antigens in a proline/threonine/serine-rich tandem repeat domain (PTS domain), thus can serve as novel therapeutic targets [230,234,235,236]. Of note, the secreted forms of those mucins, particularly MUC1, can suppress T cell proliferation and interfere with T cell activation (Figure 3a) [237,238,239].

The first generation of mAbs targeting MUC1 includes SM3, HMFG1/2 (formerly known as 1.10.F3/3.14.A3) [240,241,242], and AR20.5 [243]. SM3 recognizes both glycosylated and unglycosylated forms of MUC1 [244], and HMFG1 also shows a similar binding activity to both glycosylated and unglycosylated MUC1 [245]. However, AR20.5 has a striking binding to Tn^+^MUC1 glycopeptide over unglycosylated MUC1 peptide and exhibits CDC/ADCC activity in vitro [246]. Another feature of AR20.5 is to form immune complexes with circulating MUC1 and/or MUC1-expressing tumor cells in vivo, implying that it may stimulate a dendric cell-mediated T cell response [247]. Recently, those SM3 and AR20.5-based immunotherapies in conjugation with immunotoxin (mAb-conjugated with Granulysis, GRNLY) have been tested in Tn^+^ cancer cell lines including a pancreatic cell line, Capan-2, and a leukemic cell line, Jurkat, in both in vitro and in vivo mouse models, and resulted in targeted lysis of tumors with Tn^+^MUC1 [248]. Another novel mAb targeting MUC1 is PankoMab-GEX, which is a well-defined mAb from screening with a synthetic Tn^+^ glycopeptides [249]. Its humanized and glycoengineered version, Gatipotuzumab, was tested in patients with Tn^+^MUC1 positive tumors in Phase I and II trials, and showed well-tolerance with common adverse events (NCT01222624, and NCT01899599) [250,251]. One promising CART therapy targeting Tn^+^MUC1 was developed by the Posey group in 2016 [107], validating the concept of immunotherapy with SM3 and HMFG2 since early 2000s [252]. Consequently, HMFG2-, and SM3-scFv have been applied in the latest platform of CART technologies (HMFG2 in a third generation of CART [253]; SM3 in a PD-1 knockout CART), and they are currently being tested in clinical trials (NCT03198052, and NCT03525782, respectively).

In targeting MUC16 (Figure 3b), one of the well-characterized mAbs is 4H11 that recognizes the epitope within the retained MUC16 ectodomain next to shedding domain [254]. The MUC16-CART was developed by the Brentjens group in 2010 [255]. Later, they developed other versions of MUC16-CART with additional modifications, such as 4H11-scFv-fused with IL12, or fused with PD-1 blocking scFv, which have shown a robust anti-tumoral activity to ovarian cancer cell line in preclinical studies [256,257]. In next-generation immunotherapy, anti-MUC16 scFv (4H11-scFv) was employed on a bispecific T-cell engagers (BiTEs), which displayed redirected T-cell immunologic synapses and suppressed metastatic potential of epithelial ovarian cancer in an in vivo mouse model [258]. In enhancing anti-tumoral efficacy of MUC16-BiTEs, co-treatment with VEGF or PD-1 blocking antibodies significantly decreased tumor burden in an in vivo mouse model, suggesting that combination treatment with those blocking antibodies would be more beneficial to patients with ovarian cancer under the immunosuppressive TME. DMUC5754A, an ADC that contains a humanized anti-MUC16 with MMAE, was developed and tested in patients with ovarian and pancreatic cancers in Phase I trial, showing that DMUC5754A has anti-tumor activity in MUC16^+^ patients with an acceptable safety profile (NCT01335958) [259]. This humanized anti-MUC16 mAb was originally derived from mouse hybridoma clone 3A5 that targets multiple sites on mucin repeat of a shedding form of MUC16 [260]. Lastly, Oregovomab (originally from B43.13 [261,262]) was tested in patients with ovarian cancer in Phase II trial (NCT01616303), resulting in an improved overall survival with induced CA125-specific T cell production [263,264] and is currently being tested in Phase III trial (NCT04498117).

Other than mucins, glycoproteins CEA (carcinoembryonic antigen, also known as CEACAM5 [265]) and PSA (prostate-specific antigen [266]) are classic serum biomarkers in a majority of carcinomas [22,26,43,50,51,160,161,267,268,269]. CEA is a GPI-anchored glycoprotein overexpressed in solid adenocarcinomas of lung, colon, intestine, stomach, pancreas, liver and cervix, and the expression levels are associated with cell adhesion, differentiation, lymph node metastasis, and poor prognosis [265,270]. Like CA 19-9, CEA has also been used in cancer screening over the decades. However, CEA levels are also reported to be elevated in non-malignancies such as peptic ulcer disease, inflammatory bowel disease (IBD), and pancreatitis. Thereby, CEA is currently only used in monitoring disease course and prognosis after surgical resection as recommended in the guidelines [271,272,273]. Nevertheless, ADC version of a humanized anti-CEACAM5 mAb (SAR408377)-conjugated with maytansinoid agent DM4 has been evaluated in a preclinical setting and is currently being tested in patients with non-small lung cancer in several clinical trials (NCT04524689, NCT4394624, and NCT02187848) [274]. The other anti-CEACAM5-based drug, ADC version of labetuzumab govitecan-conjugated with SN-38 (IMMU-130, originally from MN-14 [275,276,277]) has been tested in patients with refractory or relapsed metastatic colorectal cancer in Phase I/II trial and demonstrated a manageable safety profile and therapeutic activity (NCT01270698) [278]. A Phase II study (NCT01915472) of IMMU-130 in patients with metastatic colorectal cancer was proposed and withdrawn since no patients were enrolled in any studies.

PSA is a glycoprotein exclusively secreted by prostate carcinomas over benign prostate tissues [266,279]. From a diagnostic point of view, PSA is very sensitive, and the early diagnosis with PSA test shows >50% reduction of mortality with 95% confidence and decreased the incidence of metastasis in 55 to 75-year-old men when a routine screening test was performed [280,281]. It is the single most useful tool for the early detection of prostate cancers. While its diagnostic value is highly appreciated, little has been reported in PSA-targeted therapeutics. Nevertheless, PROSTVAC, PSA-based vaccine immunotherapy-stimulating DC-T cell axis, is underway in combination therapies such as Nivolumab (anti-PD-1 mAb), or CV301 (CEA-MUC1-TRICOM vaccine) + MSB0011359C (M7824, anti-PD-L1 x anti-TGF-β receptor II BsAb) in patients with metastatic castration-resistant prostate cancer (mCRPC) in several clinical trials (NCT029333255, NCT03315871, NCT01145508, and NCT02649855) [282,283].

### 2.5. Immunotherapies Involving in Glycobiology and Glycoimmunology

As altered glycosylation on cell surface and secretary proteins is prominent in a majority of human carcinomas, TACA-targeted immunotherapy offers new opportunities for treatment of cancers. While next-generation immunotherapies have been introduced to overcome cancers such as CART, bispecific antibody, or vaccination that directly activate T-cell dependent immune response, anti-TACA immunotherapy has entered in several clinical trials, as discussed in above sections. Also, TACAs play some role in facilitating the TME in glycan-lectin receptor interaction with immune cells [17,20,27]. Therefore, the Bertozzi group has developed a new conceptual antibody-enzyme drug, bi-sialidase fusion protein (E-602), which is trastuzumab (anti-HER2 mAb)-conjugated with neuraminidase that cleaves sialic acids on HER2^+^ cancer cells, resulting in an enhanced ADCC activity and immune cell activation upon desialylation in the TME in local tumor site in preclinical studies (Figure 4) [284,285]. E-602 is currently being tested for dose escalation and dose expansion in Phase I/II trial (NCT05259696). In another example, VISTA (V-domain immunoglobulin suppressor of T cell activation, also known as B7-H5) was found to act as a binding partner to the adhesion and co-inhibitory receptor of PSGL-1 (P-selectin glycoprotein ligand-1, expressed on T cells [286,287]) at an acidic environment. At the TME, the interaction of VISTA with PSGL-1 was engaged in T cell suppression [288]. A co-treatment with a blockage against VISTA and anti-PD-1 mAb reversed immune suppressive effects in vivo [288,289]. Interestingly, the binding of VISTA requires sulfated tyrosine and protonated histidine residue on PSGL-1 on T cells, but not SLeX that is required for the binding to P-selectin in tethering and infiltration into inflamed tissues, implying that the blockage to VISTA would not interfere with PSGL-1-P-selectin-mediated immunological activity. In addition, several types of GBRs are being investigated and under development as immune modulatory targets (Figure 4): E-selectin (Phase I/II, NCT02306291, and Phase III, NCT03616470 [290,291]), SIGLECs (SIGLEC2 (CD22), NCT01564784 [292]; SIGLEC3 (CD33), NCT00927498, approved by the FDA [293]; SIGLEC15, NCT03665285 [294]), Galectins (Galectin-1, NCT01724320; Galectin-3, NCT02117362, and NCT02575404 [295,296]), NKG2 family in preclinical studies [297,298,299], and DC-SIGN in preclinical study [300]. Thus, a series of data supports that anti-TACA therapy could potentially act as a glycoimmune checkpoint to block immunosuppressive synapses and remodel the TME in local tumor sites.

## 3. Cancer Glyco-Vaccines

Vaccinations with TACAs have been thought of as robust strategies to generate immune responses since TACAs are specifically expressed on cancer cells rather than normal tissues. Due to the low immunogenicity of carbohydrates, especially those shortened epitopes of glycans and glycolipids, several approaches have been taken to improve immunogenicity over decades [301,302,303,304]. Common observation in immune responses to carbohydrate immunogens is that antibodies and GBRs require from two to six monosaccharides within glycans as an intact epitope or determinant, resulting in the limitation to induce antibodies against single monosaccharides, such as the Tn antigen [57,305,306]. Thus, carbohydrate immunogens may mainly trigger humoral immune response (IgM subclass production), but not generate long-lasting synapse of IgG class switch in plasma/memory B cells via cellular immune response [59]. In one of the examples of lesson-learned glycoimmunology, in early 1990s, Theratope^®^, which is a vaccine with STn-KLH (keyhole limpet hemocyanin) plus Detox adjuvant, was tested in patients with metastatic breast, colorectal, or ovarian cancers in active specific immunotherapy (ASI) trial, resulting in increased titers of anti-STn^+^mucin IgG in serum, which is correlated with longer survival of patients [307,308,309]. After years of clinical trials, however, Theratope^®^ was failed in Phase III trials due to no overall benefit in time to progression (TTP) or survival (NCT00046371, and NCT00003638) [310,311]. Although Theratope^®^ harnesses protective effects against cancer patients in MHC-II T cell-dependent immune response rather than humoral response that is usually triggered by most of glycans as immunogens, one of the potential complications is due to the recruiters who express a wide range of STn expression (5 to 100%) in cancers with stomach, colon, ovary, and breast due to tumor heterogeneity, indicating that Theratope^®^ in Phase III trials could have succeeded if the STn expression levels and other clinical criteria were considered such as patients who received humoral therapy showing longer overall survival with Theratope^®^ vaccination [59].

In the vaccine design, a few studies stood out that vaccinations with fully-glycosylated MUC1 synthetic peptides (i.e., Tn^+^MUC1, or STn^+^MUC1) elicit robust antibody productions and override tolerance in a mouse study [76], indicating that the glycosylation site on peptide affects antigen uptaking, processing, and presenting to T cells during antigen presentation processes in antigen-presenting cells (APCs) [312,313,314]. These data strongly suggest that the choice of glycosylation sites on glycopeptides (and immunogenic carriers; KLH, BSA, OVA, TT, and nanoparticles) would be taken into account in next-generation design and development of fine glyco-vaccines (Figure 4).

In recent decades, several other TACA-based vaccines have also undergone and been evaluated in preclinical studies [76,315,316,317,318,319,320,321,322,323,324,325,326,327,328]. As highlighted in recent glyco-vaccines in clinical trials, the Tn^+^MUC1 vaccine-mixed with autologous DCs was tested in patients with non-metastatic castration-resistant prostate cancer (non-mCRPC) in Phase I/II trial (NCT00852007), showing significant induction of T-cell response with biological activity [329]. The fully synthetic glycopeptide MAG-Tn3/AS15 vaccine, which is composed of four arms of three Tn glycotopes-linked to the tetanus toxoid (TT)-derived peptide CD4^+^ T-cell with GSK-licensed immunostimulant (AS15), was tested in patients with localized breast cancer at high-risk of relapse in Phase I trial (NCT02364492), showing that high levels of Tn-specific antibodies are detected in all vaccinated patients [330]. Unimolecular pentavalent TACA vaccine, Globo H-GM2-STn-TF-Tn with QS-21 adjuvant, was tested in patients with breast, epithelial, ovarian, fallopian tube, or peritoneal cancers in Phase I trials (NCT00030823, and NCT01248273), demonstrating the safety and development of antibodies against multiple TACAs [331,332]. Another multivalent TACA vaccine, named P10s-conjugated PADRE (pan HLA DR-binding epitope, that activates CD4^+^ T cells), which induces antibodies against GD2 and LeY, was tested with doxorubicin, cyclophosphamide and docetaxel (or paclitaxel) combination therapies in patients with high-risk breast cancer in Phase I/II trial (NCT02229084), showing an acceptable safety and immunologically promising activity [333,334]. The vaccine therapy of SLeA-KLH with QS-21 was tested in patients with metastatic breast cancer in clinical trial (NCT00470574), resulting in anticipated immunogenicity to human breast carcinomas. Active immunotherapy with Globo H-KLH (OPT-822) was tested in patients with metastatic breast cancer in Phase II trial (NCT01516307), improving humoral immune response to Globo H [335]. A trivalent ganglioside vaccine containing GM2, GD2, and GD3 (OPT-821) in combination with oral β-glucan was tested in patients with high-risk neuroblastoma, or sarcoma in Phase I/II trials (NCT00911560, and NCT01141491), eliciting robust antibody responses with improved survival [336,337].

## 4. Conclusions and Perspectives

As tumor therapeutic targets, TACAs are: (1) more tumor specific than protein targets; (2) expressed on tumor cell surface; (3) highly abundant; and (4) significant in tumor biology, i.e., suppressing and evading immune surveillance in the TME, altering oncogenic signaling pathways, promoting tumor survival, progression, and metastasis. The major limitations for TACAs to be effective immunotherapeutic targets are their low immunogenicity and a lack of T cell engagement. Nevertheless, the progresses and developments in the fields of glyco-oncology and glycoimmunology over the past few decades have facilitated the TACA immunotherapy for cancers. Particularly, the anti-GD2 immunotherapy is a success story in this regard. As the glycoimmunology continues to evolve, our better understanding of the process, presentation of the carbohydrate antigens on immune cells, and their responses to the TACAs (especially the engagement of T cells) will lead to better designs of TACA immunogens which will not only elicit more specific mAbs with higher affinity for applications of mAb therapy and CART therapy, but also develop more effective anti-cancer glyco-vaccines. Thus, the TACA-targeted immunotherapy is expected to evolve as one of the effective anti-cancer regimens in the treatment of solid tumors in a new era.

## Figures and Tables

**Figure 1 cancers-15-03536-f001:**
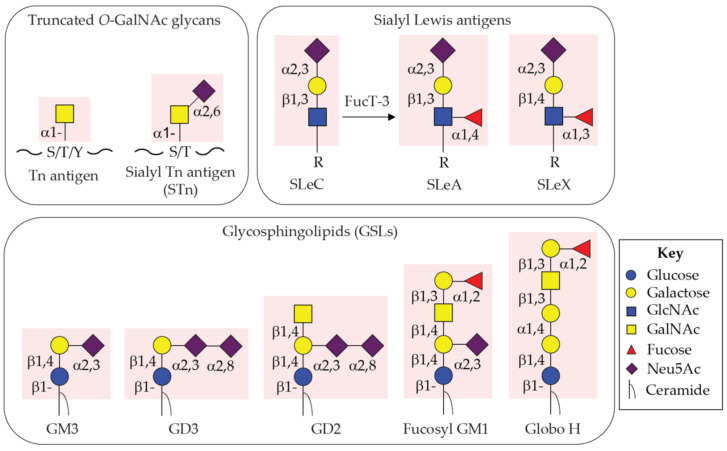
Structures of common tumor-associated carbohydrate antigens (TACAs). A cartoon depiction of the most common tumor-associated carbohydrate antigens (TACAs) in human carcinomas; *O*-GalNAc glycans (mucin-type *O*-glycans), Sialyl Lewis antigens, and Glycosphingolipids (GSLs). Highlighted in red background is antigenic determinants. S/T/Y, Serine/Threonine/Tyrosine; R, glycan moieties at the reducing end.

**Figure 2 cancers-15-03536-f002:**
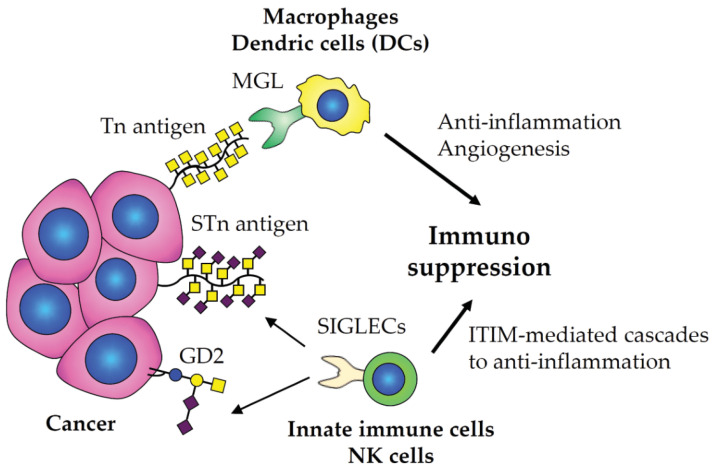
Contributions of Tn, STn antigens, and GD2 to the tumor microenvironment (TME). The Tn antigen binds to macrophage galactose-type lectin (MGL) on macrophages and dendric cells, leading to secreted anti-inflammatory cytokines, and induced angiogenesis. Sialic acid-binding immunoglobulin (Ig)-like lectins (SIGLECs) on innate immune cells and NK cells binds to sialic acids in the STn antigen and GD2 on cancer cells, inducing immunoreceptor tyrosine-based inhibitory motif (ITIM)-mediated inflammatory cascades to facilitate the TME.

**Figure 3 cancers-15-03536-f003:**
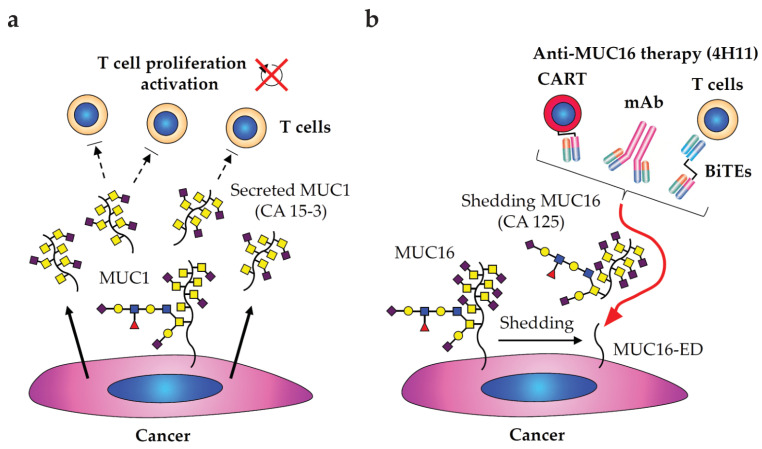
Anti-Mucin therapeutic approaches to human carcinomas. (**a**) Tumor-associated MUC1 (CA 15-3) expressed on cancer cells, or secreted from cancer cells (arrows), suppresses T cell proliferation and activation (dashed arrows). (**b**) A schematic illustration of approaches in anti-MUC16 therapy based on 4H11 that recognizes an ectodomain (ED) next to a site of shedding of MUC16 (CA 125), targeting to cancer cells (arrow in red). CART, chimeric antigen receptor T-cell, inducing CD8^+^ cytotoxicity; BiTEs, bispecific T-cell engagers (anti-MUC16 x anti-CD3 scFvs), recruiting and directly activating T cells to induce cellular immune response.

**Figure 4 cancers-15-03536-f004:**
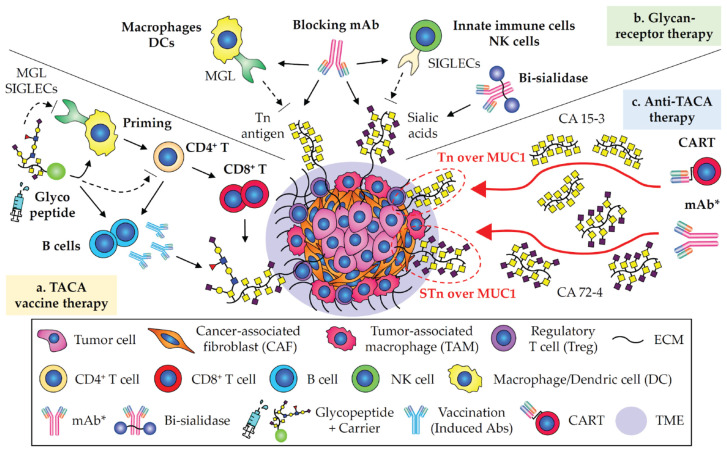
Summary of strategic approaches of anti-TACA immunotherapy in this review. (a) TACA vaccine therapy: Fine-tuning glyco-vaccine design is the key to anti-TACA vaccination. Optimizely designed glycopeptide vaccines are endocytosed, processed by antigen presenting cells (APCs), and antigenic “glyco”epitopes are presented on T cells to induce cellular immune responses (activation of CD8^+^ cytotoxic T cells, and IgG class switch in B cells) (arrows). Vaccinations with TACAs or glycopeptides should be designed not to stimulate glycan-binding receptors (GBRs) such as MGL and SIGLECs on immune cells which forge the TME (dashed arrows). Particularly, MUC1 glycopeptide vaccines with potential effects of T cell suppression should be avoided (see also Figure 3a). (b) Glycan-receptor therapy: Glycan-lectin receptor axis creates the hostile immunosuppressive TME. Blocking the binding of certain TACAs such as the Tn antigen, and sialic acid-containing glycans to their receptors, MGL and SIGLECs (arrows), could be a promising approach as a glycoimmune checkpoint (TACA-GBR axis) (dashed arrows). Bi-sialidase (E-602, trastuzumab-conjugated with sialidase) is a novel therapeutic approach to suppress the TME in local site. (**c**) Anti-TACA therapy—Consideration of the TACA targets: Although specific anti-TACA mAbs with high affinity are critical, choice of TACA targets for immunotherapy needs to be seriously considered. The potential hurdles for anti-MUC immunotherapies could arise from secreted and shedding MUCs in circulation, which might sequester CART and mAbs from tumor cells to compromise their anti-tumor activity. Furthermore, anti-MUC mAbs that target to membrane-bound mucins do not show CDC activity. Ideally, anti-MUC therapy would be targeting to MUCs that remain on cancer cell surface (see also Figure 3b). Taking that consideration, anti-Tn/STn immunotherapies targeting Tn/STn-carrying glycoproteins other than soluble MUCs may offer better options (arrows in red). *mAb includes ADC, glycoengineered version, BiTEs, and bsAb.

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
