# Peer review of "Aberrant Glycosylation as Immune Therapeutic Targets for Solid Tumors"

_cancers, 2023, doi:10.3390/cancers15143536_

Round 1

Reviewer 1 Report

This is a very well-written and timely review discussing the potential roles of aberrant cancer glycosylation as glyco-biomarkers and immune-therapy targets for solid tumors. This is one of a few manuscripts that I do not have major concerns. Please find my minor comments below.

TACAs can also be found in normal tissues to some extent. This reviewer feels fair to discuss more about physiological roles, if any, of these glycans especially in context of their therapeutic potentials in CAR-T therapy.

This reviewer wonders if “sialyl” Tn is really present on tyrosine residue. If so, please cite appropriate literatures.

The authors could mention why anti-Tn mAbs can still be valuable tools for cancer diagnosis and therapeutics, despite their apparent drawbacks, such as low immunogenicity, cross-reactivity and the presence of decoys.

Some typos need to be corrected. Line 66, relrevent; Line 78, spefic; Line 98, genrally; Line 128, infecions; Line 157, proxomicity; Line 179, plateles; Line 501, reagard; Line 502, carbihydrate. The authors should carefully check their manuscript before submitting the revised version.

Author Response

Please see attached file: "Responses to Reviewers' Comments-Cancers-Final".

Reviewer 2 Report

Summary:

In this Review article, Matsumoto & Ju summarized and discussed well-established and the most promising therapeutic targets among TACAs for monoclonal antibody (mAb) immunotherapy, antibody-drug conjugates (ADCs), chimeric antigen receptor T-cell (CART) therapy, bispecific Ab (BsAb), bispecific T-cell engagers (BiTEs), and vaccination therapy in recent clinical trials.

Comments:

The manuscript is very well written and organized into topics of great relevance to the subject addressed. The figures are very didactic, and the subtitles are self-explanatory. I congratulate the authors for the excellent review, which is certainly of interest to the audience of Cancers Journal.

Author Response

We thank the Reviewer for the encouraging comments. We appreciate your effort and time to review our manuscript.

Reviewer 3 Report

The manuscript offers a comprehensive overview of the significance and implications of aberrant glycosylation in solid tumors. It highlights the diverse nature of glycans and their crucial roles in various biological processes within normal cells. Moreover, it brings attention to the altered glycan structures found in cancer cells, known as tumor-associated carbohydrate antigens (TACAs). The authors successfully discuss the potential of TACAs as tumor biomarkers and immune therapeutic targets for the treatment of cancer.

The review provides a well-structured and organized summary of currently established and promising aberrant glycosylation targets for solid tumors. It successfully highlights the potential of targeting aberrant glycans for therapeutic interventions. Overall, the manuscript offers valuable insights into an emerging field of tumor glycobiology research with significant implications for cancer treatment.

In light of the manuscript's well-written manuscript and its relevance to the topic Glycosylation in Cancer—Biomarkers and Targeted Therapies, I am confident that this paper will be of great interest to a broad range of readers. Therefore, I recommend accepting the manuscript in its current form for publication.

Author Response

We thank the Reviewer for the encouraging comments. We agree with the Reviewer that this paper will be of great interest to the cancer research communities.